

# Conservative treatment for patients with subacromial impingement: Changes in clinical core outcomes and their relation to specific rehabilitation parameters

Mikkel B. Clausen[1,2,3], Mikas B. Merrild[1], Adam Witten[2], Karl B. Christensen[4], Mette K. Zebis[1], Per Hölmich[2] and Kristian Thorborg[2,3]

[1] Department of Physiotherapy and Occupational therapy, Metropolitan University College Copenhagen, Copenhagen, Denmark
[2] Sports Orthopedic Research Center-Copenhagen (SORC-C), Department of Orthopedic Surgery, Copenhagen University Hospital, Amager-Hvidovre, Hvidovre, Denmark
[3] Physical Medicine and Rehabilitation Research-Copenhagen (PMR-C), Copenhagen University Hospital, Amager-Hvidovre, Copenhagen, Denmark
[4] Department of Biostatistics, University of Copenhagen, Copenhagen, Denmark

## ABSTRACT

**Background**. Impaired patient-reported shoulder function and pain, external-rotation strength, abduction strength, and abduction range-of-motion (ROM) is reported in patients with subacromial impingement (SIS). However, it is unknown how much strength and ROM improves in real-life practice settings with current care. Furthermore, outcomes of treatment might depend on specific rehabilitation parameters, such as the time spent on exercises (exercise-time), number of physiotherapy sessions (physio-sessions) and number of corticosteroid injections, respectively. However, this has not previously been investigated. The purpose of this study was to describe changes in shoulder strength, ROM, patient-reported function and pain, in real-life practice settings, and explore the association between changes in clinical core outcomes and specific rehabilitation parameters.

**Methods**. Patients diagnosed with SIS at initial assessment at an outpatient hospital clinic using predefined criteria's, who had not undergone surgery after 6 months, were included in this prospective cohort study. After initial assessment (baseline), all patients underwent treatment as usual, with no interference from the investigators. The outcomes Shoulder Pain and Disability Index (SPADI:0–100), average pain (NRS:0–10), external rotation strength, abduction strength and abduction ROM, pain during each test (NRS:0–10), were collected at baseline and at six month follow-up. Amount of exercise-time, physio-sessions and steroid-injections was recorded at follow-up. Changes in outcomes were analyzed using Wilcoxon Signed-Rank test, and the corresponding effect sizes (ES) were estimated. The associations between changes in outcomes and rehabilitation parameters were explored using multiple regression analyses.

**Results**. Sixty-three patients completed both baseline and follow-up testing. Significant improvements were seen in SPADI (19 points, ES:0.53, $p < 0.001$) and all pain variables (median 1–1.5 points, ES:0.26–0.39, $p < 0.01$), but not in strength and ROM (ES:0.9–0.12, $p > 0.2$). A higher number of physio-sessions was significantly associated

Corresponding author
Mikkel B. Clausen,
mikkelbek@gmail.com

with larger improvements in external rotation strength (0.7 Newton/session, $p = 0.046$), and higher exercise-time was significantly associated with decrease in average pain ($-0.2$ points/1,000 min, $p = 0.048$).

**Discussion**. Patient-reported function and pain improved after six months of current care, but strength and ROM did not improve. This is interesting, as strengthening exercises is part of most current interventions. While two significant associations were identified between self-reported rehabilitation parameters and outcomes, the small gains per physio-session or 1,000 min of exercise-time reduces the clinical relevance of these relationships. Collectively, the findings from this study indicate room for improvement of the current rehabilitation of SIS, especially with regard to core clinical outcomes, such as strength and range of motion.

# INTRODUCTION

Subacromial impingement syndrome (SIS) is the most common shoulder disorder (*Van der Windt et al., 1995*). The condition is characterized by shoulder pain and impaired patient-reported function (*Roach et al., 1991*), but reduced strength and range of motion (ROM) has also been reported in patients with SIS (*Clausen et al., 2017*; *MacDermid et al., 2004*). More specifically, abduction and external rotation strength, as well as abduction ROM, is only two-thirds of that in the unaffected shoulder (*Clausen et al., 2017*) or healthy controls (*MacDermid et al., 2004*), which is why a $\sim$50% increase is needed to reach normal strength and ROM levels. Based on this, it is suggested that patient-reported function, pain, strength and ROM should be considered core outcomes related to SIS (*Clausen et al., 2017*).

Conservative treatment, including exercise therapy, is considered the first line of treatment (*Diercks et al., 2014*; *Sundhedsstyrelsen, 2013*), and a range of randomized controlled trials (RCTs) show promising improvements in patient-reported shoulder function, pain and ROM as a result of exercise therapy (*Akyol et al., 2012*; *Dilek et al., 2016*; *Lombardi et al., 2008*; *Başkurt et al., 2011*; *Moezy, Sepehrifar & Solayman Dodaran, 2014*; *Littlewood et al., 2015*; *Engebretsen et al., 2011*; *Yiasemides et al., 2011*; *Bal et al., 2009*; *McClure et al., 2004*; *Holmgren et al., 2012*; *Bennell et al., 2010*). However, when maximum shoulder strength is measured, improvements vary, ranging from 4–36% in external rotation strength (derived from *Dilek et al., 2016*; *Lombardi et al., 2008*; *McClure et al., 2004*; *Bennell et al., 2010*; *Maenhout et al., 2013*; *Galace de Freitas et al., 2014*; *Ingwersen et al., 2017*) and 4–42% in abduction strength (derived from *Dilek et al., 2016*; *Lombardi et al., 2008*; *McClure et al., 2004*; *Bennell et al., 2010*; *Maenhout et al., 2013*; *Galace de Freitas et al., 2014*; *Ingwersen et al., 2017*; *Struyf et al., 2013*). Importantly, however, strength improvements seem to be smallest in studies including patients whose duration of symptoms is greater than two months. In these studies, strength improvement ranges from 4% to 15% in external rotation strength and 4% to 20% in abduction strength (derived from *Lombardi et al., 2008*; *Bennell et al., 2010*; *Maenhout et al., 2013*; *Ingwersen*
*et al., 2017*). This is far from the ~50% increase needed to restore the above-mentioned shoulder strength deficits reported by *MacDermid et al. (2004)* and *Clausen et al. (2017)*. It should also be noted that both the previously reported improvements in patient-reported shoulder function, pain and ROM, as well as the less encouraging changes in strength, were all achieved in clinical trial settings. While such studies have high internal validity, it is possible that the observed improvements could partly be due to a trial effect, such as the Hawthorne effect and/or placebo effect (*Braunholtz, Edwards & Lilford, 2001*), and hence it is possible that results obtained outside such controlled settings will be less encouraging.

Conservative treatment strategies often include home-based exercises (*Holmgren et al., 2012*; *Bennell et al., 2010*). The effectiveness of such exercise interventions likely depends on patients' adherence to prescribed exercises, although this has not been demonstrated in patients with SIS. This is important because utilization of exercise therapy in the rehabilitation of patients with shoulder disorders might be limited, as indicated in a recent register-based study of 57,311 Danish shoulder patients in secondary care (70% with SIS), of whom only 43% had physiotherapy within 52 weeks after the first hospital visit, and only 62% of physiotherapy sessions included exercise therapy or advice on self-training (*Christiansen et al., 2016*). This indicates a low utilization of some specific physiotherapy and rehabilitation parameters, such as exercise therapy, potentially leading to suboptimal rehabilitation for many patients. In addition to that, the actual time that patients with SIS spend on exercise therapy has not been investigated and is therefore currently unknown. In relation to physiotherapy and exercise therapy, it is also unknown to what degree the number of completed physiotherapy sessions and the time spent on exercise therapy are related to improvements in the SIS-related core outcomes, such as shoulder strength, ROM, patient-reported function and pain (*Sundhedsstyrelsen, 2013*; *Hopman et al., 2013*).

Knowledge about improvements in SIS-related core outcomes following rehabilitation and their relationship to specific rehabilitation parameters will help researchers and clinicians understand to what degree these clinical core outcomes are addressed in current care.

### Purpose

The purposes of this study were: first, to describe the changes in shoulder core outcomes, including strength, ROM, patient-reported shoulder function and pain, from baseline to six months follow-up, in patients diagnosed with SIS at the baseline examination at a public hospital outpatient clinic; second, to describe specific rehabilitation parameters, such as the utilization of physiotherapy and the amount of home-based rehabilitation performed during the same period; third, to investigate the association between shoulder core outcomes and specific rehabilitation parameters.

## MATERIALS AND METHODS

This is a prospective cohort study based on the six-month follow-up of 129 SIS-patients (82%) from a consecutive cohort of 157 SIS-patients (*Clausen et al., 2017*). At baseline, patients underwent a clinical assessment of shoulder strength, ROM, patient-reported shoulder function and pain impairments performed by one of six trained assessors (two

physiotherapists, three physiotherapy undergraduates and one medical student). After these assessments, a clinical examination was performed by an orthopaedic surgeon shoulder specialist blinded to the results of the baseline assessments. After baseline, all patients underwent usual care in settings not provided or controlled by the investigators of this study. Accordingly, the content and progression of care was based solely on the choices of the doctors, therapists and patients, with no interference from any investigators or other intruding parties.

Approximately six months after baseline (median 28 weeks [IQR: 27; 29]), patients included in the consecutive cohort of SIS-patients were contacted by telephone, and a ten-minute interview regarding treatment and rehabilitation since baseline was conducted. Information about sick leave due to the shoulder disorder was also obtained but not presented in this study. Patients who had not undergone shoulder surgery were invited to a follow-up assessment of maximum isometric shoulder strength, ROM, patient-reported function and pain, using the same procedures as in the baseline examination. Follow-up assessments were performed by one of five assessors (physiotherapy undergraduates) who did not participate in the baseline assessment, but were trained by the primary investigator (MBC) and one of the assessors involved in the baseline assessments. Follow-up assessments were conducted at the same facility as the baseline assessments. Only the affected shoulder was tested at the follow-up assessments. The study has been evaluated by the Capitol Region Committee on Health Research Ethics in Denmark, where it was evaluated as not requiring formal ethical approval (H-3-2013-FSP29). Written informed consent was obtained from all participants.

## Participants

Inclusion and exclusion criteria for the consecutive cohort of 157 patients with SIS, who form the basis of this follow-up study, have been described in more detail in a previous paper (*Clausen et al., 2017*). Patients were considered eligible for inclusion based on the following criteria: age 18 years or more, sufficient Danish language ability, referred to examination of a shoulder problem and at least three of the five diagnostic tests for SIS were positive (Hawkins-Kennedy, Neer's, painful arc, Resisted External Rotation and Jobe's), as described by *Michener et al. (2009)*. Patients were excluded based on the following criteria: Presence of (1) a full thickness rotator cuff tear, luxation or sub-luxation of the glenohumeral or the acromioclavicular joint, frozen shoulder or osteoarthritis in the glenohumeral joint (based on clinical and/or paraclinical examination), (2) a labral lesion verified by paraclinical investigation, or (3) any competing disorder affecting the shoulder function or the ability to answer patient-reported questionnaires (e.g., neurological disease, cervical disorder, elbow disorder, mental disorder or blindness).

## Core outcome measures
### SPADI

The Shoulder Pain and Disability Index (SPADI) was used to measure patient-reported shoulder function. SPADI consists of 13 items, each scored on an 11-point numeric rating scale. The two domains pain (five questions) and disability (eight questions) are scored

separately from 0 (worst) to 100 (best), and averaged into a total SPADI score. The Danish version of SPADI is considered reliable and valid (*Christiansen, Andersen & Haahr, 2013*).

### Active abduction ROM

Active abduction ROM was measured in degrees using a digital inclinometer. First, with the subject standing with the arm in the anatomical position and the elbow extended, the inclinometer was reset on a vertical surface. The subject raised the arm in the coronal plane towards the ceiling and a measurement was taken with the inclinometer aligned parallel to the humerus, close to the insertion of the deltoid muscle. The inter-tester reliability of the test is good (ICC 0.95) (*Kolber et al., 2011*).

### Strength

Maximum isometric peak force in abduction and external rotation was assessed with the shoulder in neutral position and the elbow fully extended or flexed to 90 degrees. The subject was seated close to a wall, which was used as external resistance to the isometric contraction. Measurements were obtained using a hand-held dynamometer and measured in newtons (N). This method has been shown to have a high inter-tester reliability ($ICC_{2,1}$ > 0.9) (*Clausen et al., 2017*). Detailed descriptions of assessment procedures are freely available in the online supplement to a previous study (*Clausen et al., 2017*).

## Patient characteristics and specific rehabilitation parameters
### Demographics and disease-specific characteristics, including severity

The following disease specific characteristics were collected at baseline: Duration (duration of current shoulder problem: 0–1 months, 1–3 months, 3–6 months, >6 months); Sick Leave (on sick leave or unemployed due to shoulder problem, yes/no); Affected Side (dominant side/non-dominant side diagnosed with SIS); Age (years); and gender (male/female).

### Specific rehabilitation parameters

Information on shoulder surgery, corticosteroid injections, physiotherapy and time spent on exercises was based on the questions from the structured telephone interview (Table 1). First, patients who answered 'yes' to having had shoulder surgery (Q1) were categorised as having had surgery since baseline, and were not asked further questions regarding their treatment since baseline. The number of corticosteroid injections and physiotherapy sessions received since baseline was determined through the answer to question Q2 and Q3, respectively. Finally, total minutes of exercises was calculated as the number of weeks with exercises multiplied by the average time spent on exercises per week with exercises (Q5 ⋆ Q6).

### Global impression of change since baseline and severity at follow-up

Information on patient-reported recovery was collected from answers to the question "Is your shoulder disorder fully cured?" (yes/no). Improvement since baseline (Global impression of change, GIC) was collected through answers to the following question using a seven-point Likert scale: "How do you perceive your shoulder disorder now as compared to when you underwent the first shoulder examination six months ago?" Much worse, a very important aggravation; Worse, an important aggravation; Slightly worse,

**Table 1** Interview regarding treatment and rehabilitation since baseline.

| | Question |
|---|---|
| Shoulder surgery | Q1 "Have you had shoulder surgery since your baseline assessment six months ago?" |
| Corticosteroid injection | Q2 "Have you had a corticosteroid injection for your shoulder disorder since baseline, including the day of the baseline examination? If yes, how many times?" |
| Physiotherapy | Q3 "Have you, since the baseline examination, seen a physiotherapist for your shoulder disorder? If yes, how many times?" |
| Exercises | Q4 "Have you been doing home-based (or non-supervised) exercises for your shoulder disorder since baseline?" |
| | Q5 "For how many weeks in total since baseline have you actively been doing these exercises" |
| | Q6 "During these weeks, how much time (in minutes) did you in average use per week on the exercises?" |

but enough to be an important aggravation; The same; Slightly better, but enough to be an important improvement; Better, an important improvement; Much better, a very important improvement). The severity of the shoulder disorder was scored on a numeric rating scale (range 1–5, 1 being very mild and 5 being very severe). All information on GIC and severity was collected through a self-developed, standardized, self-administered questionnaire.

*Data reduction and statistics.* Demographics, disease specific information, baseline values for all outcomes and descriptive data on the treatment received since baseline are presented as means $\pm$ SD, median [IQR] or numbers and proportions, for conservatively-treated patients who underwent follow-up assessment and those who did not, respectively. Age was compared between groups using independent samples $t$-test; proportions were compared using $Chi^2$-test or Fisher's exact test as appropriate; baseline values, number of physiotherapy sessions and total minutes of home exercises were compared using the Mann–Whitney $U$ test.

For patients who underwent follow-up assessment, SPADI-score, strength, ROM and pain values at baseline and follow-up, as well as changes from baseline to follow up, are presented as both mean $\pm$ SD and median [IQR]. Change scores were calculated as the value at follow-up minus the value at baseline. A Wilcoxon signed-rank test was applied in order to determine whether changes were significant, and corresponding effect sizes (ES) were calculated as the test statistic divided by the number of observations ($ES = \frac{Z}{\sqrt{n}}$).

Separately for each outcome (SPADI-score, external rotation strength, abduction strength, abduction ROM, pain during each test and average pain level), the association between the change in the outcome (dependent variable) and the number of physiotherapy sessions, the time spent on home exercises and the number of steroid injections (independent variables) were investigated using multiple regression analysis. These exploratory multivariate regression analyses were performed separately for each pair

**Table 2  Baseline characteristics for conservatively treated patients, separately for those who participated in follow-up assessment and those who did not.**

| | Conservatively treated ($n = 35$) | Participated in follow-up assessment ($n = 63$) | $P =$ |
|---|---|---|---|
| Age in years, *mean ± SD* | 51 ± 15 years | 56 ± 13 years | 0.14 |
| Gender, *% females* | 51% (18 of 35) | 54% (34 of 63) | 0.84 |
| Affected side, *% dominant side* | 53% (17 of 32) | 59% (37 of 63) | 0.66 |
| Sick Leave, *% on sick leave at baseline* | 12% (4 of 33) | 8% (5 of 62) | 0.72 |
| Duration of symptoms at baseline in % | | | 0.54 |
|   *0–1 month* | 0% (0 of 34) | 3% (2 of 62) | |
|   *1–3 months* | 21% (7 of 34) | 16% (10 of 62) | |
|   *3–6 months* | 18% (6 of 34) | 26% (16 of 62) | |
|   *>6 months* | 62% (21 of 34) | 55% (34 of 62) | |
| **Baseline values** | | | |
|   SPADI Score, median [IQR] | $n = 34$  61 [43; 73] | $n = 63$  54 [38; 70] | 0.17 |
|   External rot. strength, Newton, median [IQR] | $n = 24$  59 [36; 72] | $n = 56$  52 [34; 86] | 0.77 |
|   Abduction strength, Newton, median [IQR] | $n = 25$  55 [37; 92] | $n = 56$  54 [36; 90] | 0.96 |
|   Abduction ROM, degrees, median [IQR] | $n = 27$  101 [90; 140] | $n = 58$  120 [87; 154] | 0.30 |
|   Average pain, median [IQR] | $n = 31$  4 [2.5; 5] | $n = 60$  3 [1.6; 4] | 0.02 |
|   Pain during external rot. strength, median [IQR] | $n = 24$  4 [1; 6] | $n = 56$  4 [2; 6] | 0.63 |
|   Pain during abduction strength, median [IQR] | $n = 24$  4 [0.3; 7.8] | $n = 56$  3 [1; 7] | 0.95 |
|   Pain during abduction ROM, median [IQR] | $n = 27$  5 [5; 8] | $n = 58$  5.5 [3; 8] | 0.06 |

of dependent and independent variables, including the baseline value of the dependent variable in question as covariate.

A significance level of 0.05 was applied for all statistical tests.

# RESULTS

From the original consecutive cohort of 157 patients with SIS, 129 (82%) completed the standardized telephone interview. Thirty-one of these had undergone surgery for their shoulder disorder since the baseline examination, leaving 98 conservatively-treated patients who were all invited to follow-up assessment. Sixty-three of the 98 conservatively-treated patients participated in follow-up assessment approximately six months after baseline. Demographics, disease specific information and baseline values for all outcomes are presented in Table 2, separating the conservatively-treated patients that participated in follow-up assessment ($n = 63$) from those who did not ($n = 35$). For further details on study flow, see Fig. 1.

The group of patients participating in the follow-up assessment had significantly improved from baseline to follow-up in SPADI-score (median 19 points, [3; 39], ES 0.53, $p < 0.001$) and all pain variables (median 1 to 1.5 points on NRS, ES 0.26 to 0.39, $p < 0.01$), with changes in average pain being the most pronounced. No changes were found in external rotation strength (median 2 N [−9; 17], ES 0.09, $p = 0.337$), abduction strength
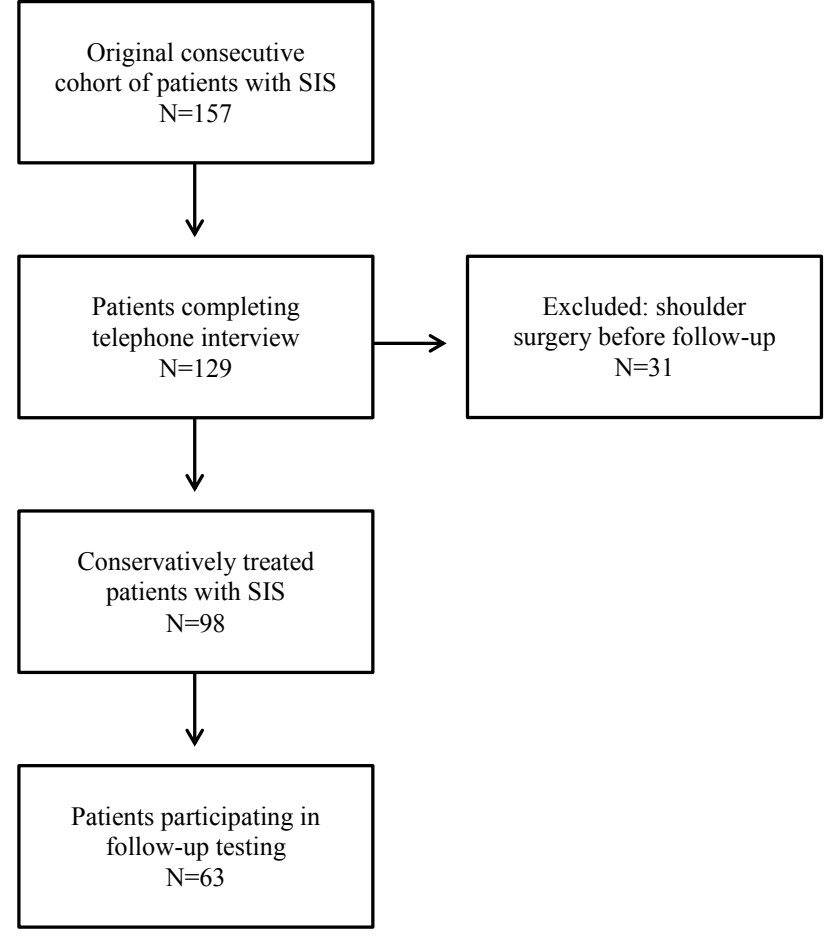

**Figure 1 Flow-chart.**

(median 4 N [−10; 16], ES 0.12, $p = 0.223$) and Abduction ROM (−1° [−12; 30], ES 0.09, $p = 0.324$). See Table 3 for further details.

At follow-up, 25% (16 of 63) rated themselves as fully recovered or much improved. The current severity of the shoulder disorder was rated ≥3 on a 1–5 scale (0 is best, 5 is worst) by 46% (29 of 63) at follow-up. For further details on global impression of change and severity at follow-up, see Figs. 2 and 3. From the conservatively-treated patients that participated in follow-up assessment 79% reported having had 0–1 steroid injections, 75% had received physiotherapy (median 5 sessions [IQR 0; 11] for the entire group) and 87% had performed exercises for their shoulder disorder (median total exercise time of 1,040 min [IQR 220; 2,700] for the entire group). The conservatively-treated patients who did not participate in follow-up assessment reported significantly lower total exercise time compared to the conservatively-treated patients who did participate in follow-up assessment ($p = 0.045$), while the other rehabilitation parameters listed in Table 4 did not differ significantly between groups. See Table 4 for further details on treatment since baseline.

**Table 3** Baseline, follow-up and change score for SPADI, strength, ROM and pain variables in conservatively treated patients who participated in follow-up assessment.

| | Outcomes | | Normality test | p-value change | Effect size |
|---|---|---|---|---|---|
| | Mean (SD) | Median [IQR] | | | |
| **SPADI score,** 0–100 points (n = 63) | | | | | |
| Baseline | 54 ± 20 | 54 [38; 70] | Normal | | |
| Follow-up | 31 ± 26 | 25 [8; 48] | Non-norm. | | |
| Change | −23 ± 24 | −19 [−39; −3] | Normal | <0.001 | −0.53 |
| **External rot. strength,** Newton (n = 56) | | | | | |
| Baseline | 62 N ± 37 | 52 N [34; 84] | Non-norm. | | |
| Follow-up | 65 N ± 35 | 56 N [44; 79] | Non-norm. | | |
| Change | 3 N ± 25 | 2 N [−9; 17] | Non-norm. | 0.337 | 0.09 |
| **Abduction strength,** Newton (n = 56) | | | | | |
| Baseline | 72 N ± 51 | 54 N [36; 89] | Non-norm. | | |
| Follow-up | 75 N ± 46 | 63 N [44; 105] | Non-norm. | | |
| Change | 4 N ± 29 | 4 N [−10; 16] | Non-norm. | 0.223 | 0.12 |
| **Abduction ROM,** degrees (n = 57) | | | | | |
| Baseline | 120° ± 40 | 121° [93; 154] | Normal | | |
| Follow-up | 128° ± 35 | 137° [107; 153] | Non-norm. | | |
| Change | 8° ± 41 | −1° [−12; 30] | Non-norm. | .324 | 0.09 |
| **Average pain,** 0–10 points (n = 60) | | | | | |
| Baseline | 3.0 ± 1.5 | 3.0 [1.8; 4] | Normal | | |
| Follow-up | 1.8 ± 2.1 | 1.0 [0; 3] | Non-norm. | | |
| Change | −1.2 ± 1.9 | −1.5 [−2.5; 0] | Normal | <0.001 | −0.39 |
| **Pain during external rot. strength,** 0–10 points (n = 56) | | | | | |
| Baseline | 4.1 ± 2.7 | 4.0 [2; 6] | Non-norm. | | |
| Follow-up | 2.6 ± 3.1 | 1.0 [0; 6] | Non-norm. | | |
| Change | −1.4 ± 2.6 | −1.0 [−3; 0] | Non-norm. | <0.001 | −0.34 |
| **Pain during abduction strength,** 0–10 points (n = 56) | | | | | |
| Baseline | 3.8 ± 2.9 | 3.0 [1; 7] | Non-norm. | | |
| Follow-up | 2.7 ± 3 | 2.0 [0; 5] | Non-norm. | | |
| Change | −1.1 ± 2.5 | −1.0 [−2; 0] | Non-norm. | 0.007 | −0.26 |
| **Pain during abduction ROM,** 0–10 points (n = 57) | | | | | |
| Baseline | 5.5 ± 2.5 | 6.0 [3; 8] | Non-norm. | | |
| Follow-up | 3.0 ± 3 | 2.0 [1; 5] | Non-norm. | | |
| Change | −2.3 ± 3.5 | −1.0 [−5; 0] | Normal | <0.001 | −0.38 |
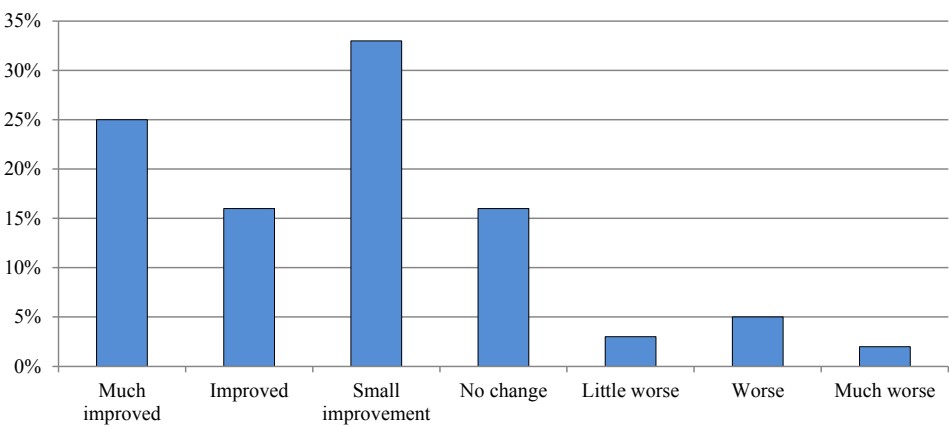

**Figure 2 Global impression of change since baseline.** Proportion of patients who participated in the follow-up assessment who reported being (1) fully cured or much improved, (2) improved, (3) small improvement, (4) no change, (5) little worse, (6) worse or (7) much worse at follow-up ($n = 63$).

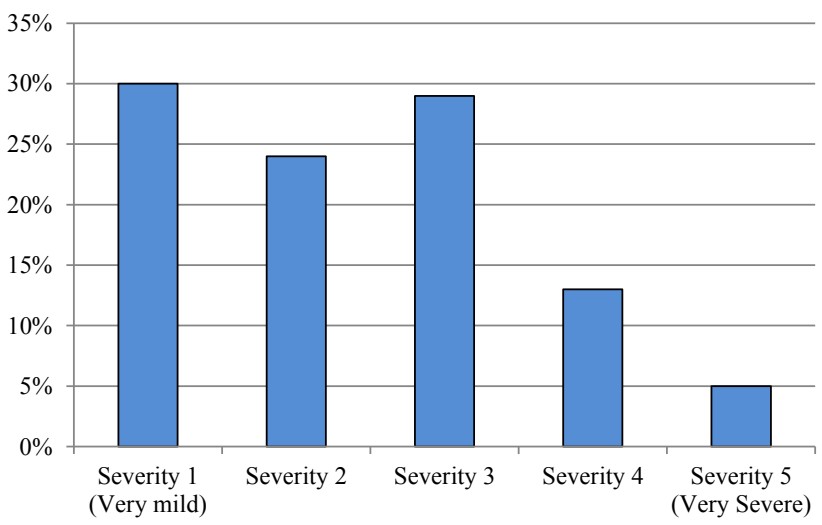

**Figure 3 Severity at follow-up.** Distribution of patient-reported severity of the shoulder disorder at follow-up, in patients who participated in the follow-up assessment ($n = 63$). Severity was scored on a 1 to 5 numeric rating scale (1 = very mild and 5 = very severe).

In the multivariate regression analyses of the results from the follow-up assessment of patients, significant positive associations were identified between the number of physiotherapy sessions and external rotation strength improvements, +0.7 N/session (95% CI [0.0–1.3], $p = 0.046$) as well as exercise-time and improvements in average pain, −0.2 points on an 11-point NRS scale per 1,000 min (95% CI [−0.4–0.0], $p = 0.048$). No additional significant associations between changes in outcome from baseline to follow-up and the total minutes of home exercises, the number of physiotherapy sessions, or the number of corticosteroid injections, were identified ($p > 0.05$), see Tables 5 and 6 for further details.

**Table 4 Specific rehabilitation parameters in the conservatively treated patients, separately for those who participated in follow-up assessment and those who did not.**

| | Not participated in follow-up assessment (n = 35) | Participated in follow-up assessment (n = 63) | P = |
|---|---|---|---|
| Number of corticosteroid, by group | | | |
| 0 injection | 46% (16 of 35) | 38% (24 of 63) | 0.16 |
| 1 injection | 31% (11 of 35) | 41% (26 of 63) | |
| 2 injections | 9% (3 of 35) | 18% (11 of 63) | |
| 3 injections | 6% (2 of 35) | 3% (2 of 63) | |
| 4 injections | 6% (2 of 35) | | |
| 5 injections | 3% (1 of 35) | | |
| Physiotherapy, %yes | 57% (20 of 35) | 75% (47 of 63) | 0.11 |
| Number of physio-sessions, median [IQR] | 2 [0; 12] | 5 [0; 11] | 0.26 |
| Grouped by number of physio-sessions | | | |
| 0 sessions | 32% (15 of 35) | 26% (16 of 62) | 0.37 |
| 1 to 5 sessions | 24% (7 of 35) | 26% (16 of 62) | |
| 6 to 10 sessions | 18% (4 of 35) | 21% (13 of 62) | |
| >10 sessions | 27% (9 of 35) | 27% (17 of 62) | |
| Home Exercises, yes n(%) | 76% (26 of 34) | 87% (55 of 63) | 0.25 |
| Total minutes of home exercises, median [IQR] | 600 [0; 1560] | 1,040 [220; 2,700] | 0.045 |

**Table 5 Regression analyses.** The influence of specific rehabilitation parameters on the change in core clinical outcomes. Adjusted for baseline value of the relevant outcome.

| | SPADI[a] (Points: 0–100) | External rot. Strength (Newton) | Abduction strength (Newton) | Abduction ROM (Degrees) |
|---|---|---|---|---|
| **Time spent on home exercises** | | | | |
| B (Δoutcome per 1.000 min) | −2.3 | 0.3 N | −0.6 N | 2° |
| (95% CI) | (−4.8 to 0.1) | (−2.1 to 2.8) | (−3.4 to 2.2) | (−2 to 5) |
| p-value | 0.058 | 0.779 | 0.654 | 0.370 |
| **Number of physio-sessions** | | | | |
| B (Δoutcome per session) | 0.3 | 0.7 N | 0.0 N | 0° |
| (95%CI) | (−0.3 to 0.8) | (0.0 to 1.3) | (−0.8 to 0.8) | (−1 to 1) |
| p-value | 0.385 | 0.046 | 0.984 | 0.962 |
| **Number of steroid injections** | | | | |
| B (Δoutcome per injection) | 4.0 | 3.8 N | 3.4 N | 0° |
| (95%CI) | (−3.0 to 11.1) | (−4.1 to 11.6) | (−5.6 to 12.4) | (−11 to 12) |
| p-value | 0.256 | 0.338 | 0.453 | 0.944 |

**Notes.**
[a]0 is best, 100 is worst. Negative change score equals improvement in symptoms.

**Table 6  Regression analyses.** Influence of specific rehabilitation parameters on the change in pain outcomes in patients participating in follow-up assessment. Adjusted for baseline value of relevant outcome.

| | Average pain during last week[a] *(0–10 points)* | Pain during tests *(NPRS, 0–10 points)* | | |
| --- | --- | --- | --- | --- |
| | | External rot. strength test[a] | Abduction strength test[a] | Abduction ROM test[a] |
| **Time spent on home exercises** | | | | |
| B (Δoutcome per 1.000 min) | −0.2 | −0.2 | +0.1 | −0.2 |
| (95%CI) | (−0.4 to 0.0) | (−0.4 to 0.1) | (−0.4 to 0.1) | (−0.6 to 0.1) |
| *p-value* | *0.048* | *0.232* | *0.236* | *0.138* |
| **Number of physio-sessions** | | | | |
| B (Δoutcome per session) | 0.0 | 0.0 | 0.0 | 0.0 |
| (95%CI) | (−0.0 to 0.1) | (0.0 to 0.1) | (−0.1 to 0.1) | (−0.1 to 0.1) |
| *p-value* | *0.726* | *0.289* | *0.651* | *0.736* |
| **Number of steroid injections** | | | | |
| B (Δoutcome per injection) | +0.5 | +0.4 | 0.0 | +0.2 |
| (95%CI) | (−0.1 to 1.0) | (−0.5 to 1.3) | (−0.8 to 0.9) | (−0.9 to 1.2) |
| *p-value* | *0.131* | *0.361* | *0.917* | *0.782* |

**Notes.**
[a] 0 is no pain, 10 is worst pain. Negative change score equals improvement in symptoms.

## DISCUSSION

In this prospective study of patients treated conservatively for SIS, medium to large effect sizes were seen for improvements in subjective outcomes of function and pain, approximately six months following initial assessment, in 63 conservatively treated SIS-patients. Interestingly, objective measures of strength and ROM did not improve.

### Changes in strength and ROM

To the best of our knowledge, it has not previously been reported to what extent shoulder strength and ROM have changed in patients with SIS who underwent conservative treatment outside a controlled clinical trial setting. Therefore, the most important finding of this study is that usual care did not improve these core clinical outcomes assessing important impairments such as strength and mobility, with non-significant median changes in glenohumeral strength and shoulder abduction ROM close to zero (−1 degree and 2 to 4 N, $p > 0.2$). This is despite the fact that shoulder mobility and strengthening exercises aimed at the rotator cuff are a part of most treatment programs (*Littlewood et al., 2015*; *Holmgren et al., 2012*; *Bennell et al., 2010*; *Struyf et al., 2013*). Such lack of improvement is especially relevant considering that the same group of patients have been shown to lack ~50% in abduction ROM as well as abduction and external rotation strength, to reach the same level as the unaffected shoulder (*Clausen et al., 2017*).

Before our study, data regarding improvements in glenohumeral strength in patients with SIS, who had received an intervention including rotator cuff strengthening exercises, have solely been available from clinical trials, revealing somewhat varying results. Accordingly, external rotation strength (*Dilek et al., 2016*; *McClure et al., 2004*; *Maenhout et al., 2013*;

*Galace de Freitas et al., 2014*; *Ingwersen et al., 2017*) and abduction strength (*Dilek et al., 2016*; *McClure et al., 2004*; *Maenhout et al., 2013*; *Ingwersen et al., 2017*) improved significantly in some of the trials, with improvements ranging from 15% (derived from *Ingwersen et al., 2017*) to 43% (derived from *Dilek et al., 2016*). In five clinical trials, strength in abduction (*Lombardi et al., 2008*; *Galace de Freitas et al., 2014*; *Struyf et al., 2013*) and external rotation (*Lombardi et al., 2008*; *Bennell et al., 2010*) did not change significantly. Only in the two trials by *Dilek et al. (2016)* and *Galace de Freitas et al. (2014)* did average improvements in abduction and/or external rotation strength exceed 20%. The trial by *Dilek et al. (2016)* in particular stands out, not only with pronounced strength improvements in both abduction (43%) and external rotation (32%), but also near-perfect outcomes in pain scores, with minimal-to-no pain at 12-weeks follow-up for patients who had received an intervention including rotator cuff strengthening exercises. While these treatment outcomes are impressive, they indicate that the population included by *Dilek et al. (2016)* might not be comparable to that of other studies, including the current study, so the encouraging results regarding increases in shoulder strength may not represent the typical response pattern. In support of Dilek et al. large improvements (36%) in external rotation strength were also found in the trial by *Galace de Freitas et al. (2014)*. In that trial, the population seems to mirror the population included in the present study and most of the aforementioned trials that included shoulder strength as an outcome (*Lombardi et al., 2008*; *McClure et al., 2004*; *Bennell et al., 2010*; *Maenhout et al., 2013*; *Ingwersen et al., 2017*; *Struyf et al., 2013*), indicating that pronounced strength improvement may be possible in these populations, but is often not achieved. In summary, based on the findings from our study and previous studies, it appears that current treatment strategies in the available scientific literature often do not address shoulder strength impairments sufficiently in patients with SIS, and future research should aim to improve the rehabilitation of these impairments.

The non-existent change in abduction ROM between baseline and follow-up ($-1°$ [IQR $-12$; 30], $p = 0.324$) in this study, with the median at follow-up reaching only 137° [IQR 107; 153], is in contrast to the more encouraging results found in patients with SIS who underwent active treatment with shoulder exercises in clinical trial settings. In these trials, improvements of >20 degrees (*Akyol et al., 2012*; *Lombardi et al., 2008*) or follow-up results close to 180 degrees (*Dilek et al., 2016*; *Başkurt et al., 2011*) were reported for groups with varying starting levels of abduction ROM. Interestingly, the lack of improvement in abduction ROM found in our study is comparable to the treatment results in control group patients on the waiting list for two months in the RCT study by *Lombardi et al. (2008)*. Accordingly, it seems that the results of current care, regarding the core outcome of shoulder abduction ROM, are a long way from matching the encouraging results found in clinical trials. While this does question the effectiveness of current care in the rehabilitation of patients with SIS, it also indicates an important difference in treatment responses obtained in clinical trials compared to real life settings. This apparent difference in treatment response underlines the importance of conducting RCTs using a more pragmatic approach to obtain valid data on the target group regarding the effectiveness

of rehabilitation interventions. Such an approach would help inform both clinicians and policy makers when recommending or implementing specific rehabilitation strategies.

## Changes in patient-reported function, pain, global impression of change and severity

In contrast to the objective outcomes of shoulder strength and ROM, both patient-reported function and all pain outcomes improved significantly in the current study (mean SPADI improvement of 23 points, median pain improvements of 1–1.5 points on 11-point NRS). This improvement in SPADI score is comparable to the mean improvement of 22.4 points after 22 weeks reported by *Bennell et al. (2010)*, the 23.5 and 29.1 points (control and intervention group, respectively) after six months reported by *Littlewood et al. (2015)*, and the 24.8 points after one year reported by *Engebretsen et al. (2011)*, all in comparable populations. It is, however, smaller than the 32.7 and 37.2 points (control and intervention group, respectively) after 12 weeks reported by *Bal et al. (2009)*. This dissimilarity to the current and previous studies (*Littlewood et al., 2015*; *Engebretsen et al., 2011*; *Bennell et al., 2010*) could be explained by a seemingly shorter symptom duration in the population studied by *Bal et al. (2009)*, as compared to our study and previous studies (*Littlewood et al., 2015*; *Engebretsen et al., 2011*; *Bennell et al., 2010*). In summary, the results of conservative treatment in this study are fairly encouraging, with mean improvements in SPADI scores that are comparable to those found in previous trials (*Littlewood et al., 2015*; *Engebretsen et al., 2011*; *Bennell et al., 2010*). This is, however, in slight contrast to the fact that only 25% of the patients considered themselves fully recovered or much improved at follow-up and median follow-up SPADI scores of 25 points [IQR: 8; 48], revealing room for improvement in current care.

## Objective versus subjective outcomes

In this study of patients with SIS who underwent conservative treatment in secondary care, patient-reported outcomes improved significantly while objective outcomes did not. Considering the lack of relationship between objective and subjective outcomes in this population (*Clausen et al., 2017*), this inconsistency might not be surprising. However, these findings further suggests that the focus of treatment has, rightfully, been on the subjective symptoms, which are considered cardinal in SIS (*Roach et al., 1991*), while strength improvements have not necessarily been the focus of rehabilitation. In addition, the observed inconsistency is also likely affected by the larger effect of the non-specific parts of treatment (i.e., placebo) on subjective outcomes compared to objective outcomes, as has previously been described by *Hróbjartsson & Gøtzsche (2001)*. This, combined with the knowledge that objective impairment measures and subjective outcomes seems to measure different constructs (*Clausen et al., 2017*), underlines the importance of including both when evaluating treatment effect.

## Rehabilitation parameters and relation to outcomes

The relationship between specific rehabilitation parameters and the clinical outcomes of treatment was limited in the current study, indicating that core clinical outcomes are not sufficiently addressed in current care. Accordingly, only two statistically significant

associations were identified, with the effects $+0.7$ N increase in external rotation strength per physiotherapy session (95% CI [0.0–1.3], $p = 0.046$) and $-0.2$ points decrease on 0–10 NRS per 1,000 min of patient-reported exercise time ($-0.4$ to $0.0$, $p = 0.048$). In this context, it should be noted that the large number of regression analyses performed increases the risk of a type 1 error. In fact, adjusting for this using a Bonferroni correction would mean that no relationships would come out as significant. Therefore, considering the limited gains per session or 1,000 min, the borderline significance of these associations and the risk of type 1 errors, the relationship between the specific rehabilitation parameters investigated in the current study and the clinical outcomes cannot be considered clinically relevant.

## Strengths and limitations

An important strength to the current study is the application of a consecutive sampling strategy, which increases the generalizability of the results. It should be noted, however, that one third of the conservatively-treated SIS-patients did not participate in follow-up assessment of strength and ROM. This might reduce the internal validity of the results due to selection bias in case of systematic differences between patients who did and did not participate in follow-up assessment. However, when comparing baseline and rehabilitation parameters, we found that only average pain at baseline ($p = 0.02$, see Table 2) and the total time spent on exercises ($p = 0.045$, see Table 4) differed significantly between groups, indicating that the risk of selection bias is minor, though it should still be considered when interpreting the findings of the current study.

In the regression analyses, we only adjusted for baseline scores of the relevant variable, and not for other covariates, such as duration of symptoms. While this leaves a risk of residual confounding, *post hoc* analyses including duration of symptoms as a covariate in all regression analyses did not have a relevant impact on the results, clearly indicating that symptom duration did not confound the relationship between rehabilitation parameters and outcomes in this study. For the variables *time spent on exercises* and *number of physiotherapy sessions*, only exercise time and number of sessions were recorded. Therefore, one could argue that important differences in the content of exercises and physiotherapy sessions have been ignored, as such information would provide additional insight into the relationships between treatment and outcome. There is also a risk of recall bias, especially related to the variable *time spent on exercises*. As studies have shown that, even in exercise diaries, patients vastly overestimate the amount of exercise they perform (*Rathleff et al., 2016*), this will in turn decrease the improvement in outcomes per reported exercise time. However, looking from a patient perspective, it seems relevant that the more time you put into performing exercises or the more physiotherapy sessions you attend, the better the outcome, at least when focusing on main outcomes such as SPADI and pain. Nevertheless, based on the inherent limitations to the applied method, it should be stressed that the exploratory finding of no relevant relationship between patient-reported specific rehabilitation parameters and treatment outcomes should be interpreted with caution. Rather, the results presented here serve as a clear indication that more research is needed to

investigate how to address specific clinical outcomes, and to what extent these outcomes are addressed in current care, using prospectively recorded and valid measures of adherence to specific exercises and the content of physiotherapy.

## CONCLUSION

In conclusion, conservatively-treated patients with SIS did not improve in objective clinical outcomes, but significant improvements were seen in patient-reported outcomes. Furthermore, no relevant relationships between specific rehabilitation parameters and improvements in outcomes were identified, but these findings should be interpreted with caution. Collectively, however, the findings of the current study indicate room for improvement of the current rehabilitation of SIS, especially with regard to core clinical outcomes, such as strength and range of motion.

## ACKNOWLEDGEMENTS

The authors would like to thank the patients who participated in this study, and all the personnel at the Sports Orthopedic Research Center-Copenhagen (SORC-C) for their assistance.

### Funding

The study was funded by the Danish Ministry of Higher Education and Science (via the Metropolitan University College); Sports Orthopedic Research Center-Copenhagen (SORC-C), Department of Orthopedic Surgery, Copenhagen University Hospital, Amager-Hvidovre, Denmark and Praksisfonden (15/808). The funders had no role in study design, data collection and analysis, decision to publish, or preparation of the manuscript.

### Grant Disclosures

The following grant information was disclosed by the authors:
Danish Ministry of Higher Education and Science.
Sports Orthopedic Research Center-Copenhagen (SORC-C).
Department of Orthopedic Surgery, Copenhagen University Hospital, Amager-Hvidovre, Denmark and Praksisfonden: 15/808.

### Competing Interests

The authors declare there are no competing interests.

### Author Contributions

- Mikkel B. Clausen conceived and designed the experiments, performed the experiments, analyzed the data, contributed reagents/materials/analysis tools, prepared figures and/or tables, authored or reviewed drafts of the paper, approved the final draft.
- Mikas B. Merrild and Adam Witten performed the experiments, authored or reviewed drafts of the paper, approved the final draft.
- Karl B. Christensen analyzed the data, authored or reviewed drafts of the paper, approved the final draft.
- Mette K. Zebis and Kristian Thorborg conceived and designed the experiments, authored or reviewed drafts of the paper, approved the final draft.
- Per Hölmich conceived and designed the experiments, contributed reagents/materials/-analysis tools, authored or reviewed drafts of the paper, approved the final draft.

## Human Ethics

The following information was supplied relating to ethical approvals (i.e., approving body and any reference numbers):

The study has been evaluated by the Capitol Region Committee on Health Research Ethics in Denmark, where it was evaluated as not requiring formal ethical approval.

## Data Availability

The raw data is available as a Supplemental File.

## Supplemental Information

Supplemental information for this article can be found online at http://dx.doi.org/10.7717/peerj.4400#supplemental-information.

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
