# Peer review of "Conservative treatment for patients with subacromial impingement: Changes in clinical core outcomes and their relation to specific rehabilitation parameters"

_PeerJ, doi:10.7717/peerj.4400_

## Round 0.1 · original submission · Major Revisions

Thank you again for your submission. I am intrigued by your study; however, there are quite a few concerns that would need to be addressed upon resubmission. Please read, review, and respond to each of the reviewers' comments in detail. In particular, it will be very important for you to describe the typical exercises/rehab protocols used. While I understand this may be quite variable, you will need to show some more detailed description of these. Also, please note that Reviewer 2 has attached a separate document with further comments/suggestions to which you should respond.

I look forward to your resubmission.
Scotty

·

Basic reporting

First of all, the trial evaluates an interesting research question. Its design is be able to evaluate the basic research questions the authors are trying to answer. However, several limitations make the trials results partly invalid.

In general the manuscript is acceptably well written. However, as I am not native English speaking, I would recommend review from a native on this point, as some phrases could be optimized.

In line 28+39, a statement regarding 50% lack in strenght is reported for SIS patients based on ref 3 and 4. When reading reference 3, and 4, I can´t find where the 50% is reported.In general I can only find defiencies up to 37% in ref 4 and 33% in ref 3. Furthermore, reference 4 is based on a biased population of healthy participants working in a healthcare setting, and thereby not equal to a “normal” patient population. As this is a primary argument throughout the article, that patients lack 50% strength in general, and often minor strength gains are obtained in clinical trials, I worry the rationale for the discussion is screwed towards questioning the effect in clinical practice based on a unvalid claim.

Line 164 delete “AND” in “recovery and was collected…”

Experimental design

Line 86-88: I suspect that 10 min. interviewes performed 6 months post baseline, regarding amount of physiotherapy sessions and time spent on exercise, is highly influenced by recall bias. Especially as most patients will have been treated by physiotherapy closer to the baseline measurement then the follow-up time. It could be suspected that in many cases a 3-4 months recall period may influence the data. This should be considered a major limitation of the study results, probably overestimating the number of sessions/minutes performed for gains in function, as patients will tend to overestimate the amount of exercise performed and number of treatments by physiotherapy. This is discussed in the article, however, the potential influence on the results could to be furthere discussed in the article.

Line 92-93: It seems that a new group of testers performed the follow-up testing. Even though the authors present their methods for strength and ROM measurements as being reliable, it would be expected that some inter rater variation could have been introduced. Especially as the raters are students. The authors describe that all raters have trained with the PI. However, I wonder why the healthy arm have not been measured at the follow-up also, in order to minimize this effect, by looking at between arm differences.

Validity of the findings

I am not a statistical expert, but I was wondering if the multiple regression model described in line 193-199 is the optimal method for answering if there is an association between nr of sessions /time spent on exercise results and effect on PRO/Strength/ROM/Pain. It could be speculated that chronicity could have an impact, and maybe duration should be included as a covariate. A scatterplot of change X time/nr of sessions would be helpfull to support the findings from these multiple regressions.

Line 200: A significance level of 0.05 is used. However with the amount of different tests, an ”0.05/number of analysis performed” correction (Bonferroni correction) could be applied, in order to minimize the risk of finding a statistical significant effect by chance.

Line 203-204: What is the reason for not including all 157 patients from the original consecutive cohort?

line 206: The authors report that 63% of the population is participating in the follow-up. This is 63% of the 98 patients treated conservatively. Why not present the actual percentage the sample consist of from the included patients (48%).

Line 290-294: I find it a bit one-sided only to discuss the lack of strength/ROM gain to be an effect of placebo. Several other causes could influence this (e.g. indurance training VS strength training; Remaining amount of pain; Chronicity).

Line 305-316: A 19 points change in SPADI – how is this somewhat disappointing. Several other factors could have influenced this trial results (e.g type of training/amount of training/degree of chronicity in the population) to expect a change in strength. I would recommend minimizing the focus on the lack in strength gains as a consequence of the physiotherapy intervention, when there are so many uncontrolled reasons for why the small population included hasn’t changed. Physiotherapy rehab is much more than giving strengthening exercises – maybe a major proportion have been instructed in performing less work with their shoulder in general, to decrease a potential overloaded shoulder and so on.

I lack an explanation for why the entire group has not been included from the original cohort. When presenting the results from the patients undergoing the telephone interview and the group participating in the follow-up tests, I lack a separate presentation on the group of patient declining from participation in the follow-up (35 patients treated conservatively) – the way the groups are presented now makes it difficult to confirm the statement in line 322-326.

Line 348-349: In general this conclusion is not well supported by this trial, based on the issues mentioned earlier.

Additional comments

Line 239-240: Presenting it as “Surprisingly” that no change is seen in the objective measures is maybe an overstatement. In the trial population, a large percentage does not recover completely, and as long as there is some degree of pain, a significant change in especially strength cannot be expected, as pain will inhibit the MVC. Furthermore, the exercises performed in rehabilitation cannot be expected to focus on strength gain. The authors could maybe have looked at endurance instead of max strength.

Line 251: 50% is used as percentage on how much patients in general lack in strength/ROM. As mentioned previously, I can’t see this statement is not supported by the reference used.

Line 259-261: I would suggest to delete this phrase, as the information is included in the sentence line 255-257.

I would recommend to change the wording to “…findings from ours and some of the previous studies…”

Line 333-336: Please consider to rephrase – I don´t completely understand the point trying to be made.

Reviewer 2 ·

Basic reporting

Not clear in many areas.
See separate comment sheet.

Experimental design

Methods need more detail.
Literature review on previous clinical trials looking at strength and needs to be more detailed.
See separate comment sheet.

Validity of the findings

Data collected was lacking key information to make the types of statements made about conservative treatment weak.
See separate comment sheet.

Additional comments

Overall, I can see the value and importance of their main research questions. However, I think a lot revisions are needed. The language was not clear and would benefit from a native English speaker to help with awkward phrases and word choices. I tried to assist with providing examples.

Methods need to be clearer. Referencing a methodology used in a previous paper may be acceptable but not on key pieces of information that define the patient population being studied. When making comparisons to previous clinical trials and their data, and when discussing the variability of findings from previous trials, the authors need to present a stronger case by outlining the types of reviews these were (design, sample size, perhaps provide a summary table). This would strengthen their paper.

I thought they needed to capture more details on the types of treatments their patients received for the SIS. The number of hours and sessions doesn't tell you anything about the quality of the treatment and content. Exercises can mean a huge range of things and so capturing time doing "exercise" can't be translated to all types of practices. They say that the nature of exercises likely differs between subjects but there is a huge range in what "exercise" could be defined as by a patient's perspective and also by therapists level of training, experience, practice setting etc.

The generalizability of their findings is limited.

Annotated reviews are not available for download in order to protect the identity of reviewers who chose to remain anonymous.

---

## Round 0.2 · Minor Revisions

Hello - thank you for your revised manuscript, which is much improved, and your attention to the reviewer's comments. There are a few points of clarification requested by Reviewer 1 that will help clarify a few things. Please address these comments in your next submission.

Scotty

·

Basic reporting

Table 2:
Title: ”….Assessment those who did not”. I would recommend to add ”And” between assessment/Those.

In general it can be misunderstood when the group who did not participate in follow-up is called “Conservatively treated” – it makes you wonder what the other group (who participated in follow-up) received. Use the same label as in table 4.

Question: “Duration at baseline” – Duration of what?

Table 4:
There is a mismatch between the n in “physiotherapy, %yes” and “Grouped by number of Physio-Sessions” for the “Participants in follow-up assessment” answering “0 Sessions” (16 answering “NO” to physio and 17 answering “0” Physio sessions)
Baseline values: “n=XX” is shifted so the numbers is on two lines.

Experimental design

No comments

Validity of the findings

Answer to my precious "Comment 13":
I would recommend adding that patients declining to participate in follow-up test, probably/might are those who have had the largest positive change. This means that, the significantly less time they have exercised would have an impact on the regressions analysis’s, and this should be pointed out in the discussion section, and not just be rejected as not influencing the results or the validity of the results. (Line 358 to 361 in revised manuscript)

Additional comments

Thank you for considering my previous comments, and acting accordingly. I have no more comments to the manuscripts then the above mentioned.

Reviewer 2 ·

Basic reporting

No comment

Experimental design

No comment

Validity of the findings

No comment

Additional comments

I had a chance to review this manuscript initially and it has improved sufficiently since then. Most of my concerns were brought to the authors' attention through reviewer #1. I feel the authors have sufficiently address the concerns.

---

## Round 0.3 · accepted · Accept

Thank you for the fast turn around with these most recent minor revisions. I am satisfied with your responses, corrections, and rebuttals. Congratulations on a great manuscript!

Scotty